# Resource Allocation and Sharing Methodologies When Reconfigurable Intelligent Surfaces Meet Multiple Base Stations

**DOI:** 10.3390/s22155619

**Published:** 2022-07-27

**Authors:** Yoghitha Ramamoorthi, Riku Ohmiya, Masashi Iwabuchi, Tomoaki Ogawa, Yasushi Takatori

**Affiliations:** NTT Access Network Service Systems Laboratories, Nippon Telegraph and Telephone Corporation, Yokosuka 239-0847, Japan; riku.oomiya.ez@hco.ntt.co.jp (R.O.); masashi.iwabuchi.vs@hco.ntt.co.jp (M.I.); tomoaki.ogawa.yg@hco.ntt.co.jp (T.O.); yasushi.takatori.rk@hco.ntt.co.jp (Y.T.)

**Keywords:** 6G, resource allocation, reconfigurable intelligent surface (RIS), RIS elements, scheduling, sharing, time

## Abstract

The 6G wireless systems are expected to have higher capacity, reliability, and energy efficiency than the existing cellular systems. Millimeter-wave (mmWave) frequencies offer high capacity at the cost of high attenuation and blockage losses. Reconfigurable intelligent surface (RIS) assisted mmWave networks consist of smaller antenna elements that control the propagation channel between the base station (BS) and the user by appropriately tuning the phase and the reflection of the incident electromagnetic signal. The deployment of RIS is considered to be an energy efficient solution to improve the coverage of regions with high blocking probability. However, if every BS is associated with one or more dedicated RIS, then the density of RIS increases proportionally with the density of BSs. Hence in this work, we propose RIS sharing mechanisms where multiple BSs share one RIS. We formulate resource allocation of RIS sharing in terms of time and the RIS elements as an optimization problem, and we propose heuristics to solve both. Further, we present detailed simulation results to compare time and the element based RIS sharing methods for various scenarios with the benchmark and the RIS system without sharing. The proposed time and element based RIS sharing methods improve throughput upto 53% and 25%, respectively, compared to the RIS system without sharing in specific scenarios.

## 1. Introduction

The next generation 6G networks are anticipated to have high reliability, capacity, and energy efficient architectures. One of the primary ways to improve the capacity is to operate at high frequencies. The deployment and the use of millimeter wave (mmWave) frequencies are evolving with incredible speed. The mmWave offers high bandwidth at the expense of high path loss due to higher attenuation over a smaller distance [1]. The intelligent reflecting surfaces are introduced to control the propagation channel by properly tuning the phase and the amplitude of the reflected signal. The reconfigurable intelligent surfaces (RIS), broadly known as an intelligent reflecting surface, consists of the smaller antenna elements capable of reflecting the information signal with a certain phase [1]. The RIS actively controls the phase and reflection of the signal using its antenna elements, thus making the wireless channel controllable.

The RIS-aided multi-user networks with RIS sharing have been achieved through RIS hybrid beamforming in [2]. A flexible time sharing non-orthogonal multiple access (NOMA) allowing users to divide their data into parts that are transmitted via NOMA and time division multiple access (TDMA) has been discussed in [3]. The joint design of transmit power allocation and the RIS reflection coefficients based on the channel estimated through grouping RIS antenna elements has been proposed in [4]. The deployment of RIS assisting two heterogeneous communication services has been presented in [5]. The joint design of time allocation, cooperative beamforming, and the reflection coefficients for RIS aided cellular and wireless powered sensor networks have been considered in [6]. The RIS with its elements grouped in blocks assisting multiple users has been formulated, and two heuristics have been proposed to solve them in [7]. The user association strategy with the user mobility for RIS-aided multi-beam transmission systems have been considered in [8], and the ON/OFF selection of RIS in a massive RIS aided wireless communications has been proposed in [9]. The resource allocation of RIS-aided dual connectivity has been presented in [10]. The prospective use cases of 6G includes RIS assisting wireless networks [1], NOMA [3], edge computing [11], etc., as in [1,2,3,4,5,6,7,8,9,10,11].

Most of the literature in the RIS aided wireless networks has been focusing on the joint resource allocation of time, power, frequency, and reflection coefficients. The RIS considered in those studies has been associated with a single BS or jointly served by multiple BSs focusing towards a single user. However, how these RIS resources are allocated to multiple BSs in proximity has not been considered yet. To the best of our knowledge, this is the first work that considers the resource allocation of RIS in terms of time and frequency when shared among multiple BSs. Considering RIS is an energy efficient relaying method to improve coverage, RIS sharing can eliminate interference by effectively utilizing it among BSs in terms of time and RIS elements. With this motivation, the contributions made in this work are as follows. The resource allocation of time and element based RIS sharing among multiple BSs are formulated as an optimization problem for an α−fair scheduler. Two heuristics are proposed to solve the formulated time- and element-based RIS sharing problem. Further, the detailed simulation results of the proposed sharing methods are presented for various scenarios that show a significant improvement in coverage and throughput.

The system model considered in this paper is presented in Section 2. The formulated problem and the associated heuristics of time and element based RIS Sharing are presented in Section 3 and Section 4, respectively. The results of the numerical evaluation are presented in Section 5. The conclusion and the future works are given in Section 6. The system model is presented next.

## 2. System Model

We consider orthogonal frequency division multiple access based cellular system. We define J = {1,2,3,…,J}, U = {1,2,3,…,U}, and R = {1,2,3,…,r} as the set of BSs, users, and RISs in the system, respectively. The set of elements in any RIS is denoted by N = {1,2,3,…,N}. The cellular system with a RIS shared between BS1 and BS2 with their users is illustrated in Figure 1. The mathematical notations are presented in Table 1. The physical channel modeling is presented in next subsection.

### 2.1. Physical Channel Model

We focus on the downlink (DL) in a time division duplexing (TDD) based system. Let $h_{u,j}^{0}$ and $x_{u,j}^{0}$ denote the direct channel gain and the binary association between BS *j* and the user *u*, respectively. $x_{u,j}^{0}$ is 1 if the direct link exists between BS *j* and the user *u* and 0 if it is not. We consider maximum received power-based association, and it is given as follows.
(1)$$\begin{array}{c} x_{u,j}^{0}=\left\{\begin{array}{c} 1,\mathrm{if}\;j=\arg\max_{j}\left\{P_{j}h_{u,j}^{0}\right\},\\ 0,\mathrm{otherwise},\forall u\in U,\;\forall j\in J,\; \end{array}\right. \end{array}$$
where, $P_{j}$ is the transmit power of BS *j* and $h_{u,j}^{0}$ is the direct link channel gain between BS *j* and the user *u* as expressed below.
(2)$$h_{u,j}^{0}=10^{-\left(\frac{PL\left(u,j\right)+\upsilon +\xi -G_{j}-G_{u}}{10}\right)},$$
where, PL(u,j) is the pathloss between user *u* and BS *j*, υ is the small scale fading loss, ξ is the loss due to log-normal shadowing, $G_{j}$ is the transmit antenna gain, and $G_{u}$ is the gain of user antenna. The DL signal-to-interference-plus-noise (SINR) ratio of a direct link between BS *j* and the user *u* at the user is written as
(3)$$\omega_{u,j}^{0}=\frac{P_{j}^{c}h_{u,j}^{0}}{\sum_{j^{\prime }\in J\backslash j}P_{j^{\prime }}^{c}h_{u,j^{\prime }}^{0}+\sigma^{2}},\;\forall u\in U,\;$$
where, $\sum_{j^{\prime }\in B\backslash b}P_{j^{\prime }}^{0}h_{u,j^{\prime }}^{0}$ is the interference from the other BSs, $P_{j}$ is the transmit power of BS *j*, and $\sigma^{2}$ is the noise power. The channel model and the scheduling of RIS are presented next.

### 2.2. RIS Channel Model and Scheduling

When there is no possibility of a direct link between the BS *j* and the user *u*, i.e., if the user is in non-line of sight (NLOS) region of the BS, then the RIS channel is said to be accessible. The RIS based physical channel comprises two cascaded channel links. One is the channel between BS *j* and RIS *r* denoted as $y_{j}^{r,n}$ and the other one is between RIS *r* and user *u* denoted as $z_{u,j}^{n,r}$. The superscript *n* implies the nth reflective element. The overall channel between user *u* and BS, *j* via RIS *r*, is the cascading of the above two channels along with the reflective parameter matrix of RIS (Θ), i.e., $y_{j}^{r,n}\Theta z_{u,j}^{r,n}$. The SINR of user *u* from BS *j* via RIS *r* is denoted by ${\hat{\omega }}_{u,j}^{r}$ as given below
(4)$$\omega_{u,j}^{r}=\frac{P_{j}\left(h_{u,j}^{0}+y_{j}^{r,n}\Theta z_{u,j}^{r,n}\right)}{\sum_{j^{\prime }\in J\backslash j}P_{j^{\prime }}h_{u,j^{\prime }}^{0}+\sigma^{2}},\;\forall u\in U,\;\forall r\geq 1,\;$$
where, $h_{u,j}^{0}$ is the direct channel gain between BS *j* and user *u*, Θ is the diagonal matrix containing phase shifts ($\theta_{r,n}$) and reflection coefficient values ($\beta_{r,n}$) of each element *n* of RIS *r*. The $\sum_{j^{\prime }\in B\backslash b}P_{j^{\prime }}^{0}h_{u,j^{\prime }}^{0}$ is the interference from the other BSs, $P_{j}$ is the transmit power of BS *j*, and $\sigma^{2}$ is the noise power. The interference from the other RIS is negligible because the received signal is assumed to be coherently combined and beamformed towards the intended receiver *u*. Please note we consider interference from other BSs in this work. Given $\omega_{u,j}^{r}$, the binary user association between BS *j* and the user *j* via RIS *r* is denoted by $x_{u,j}^{r}$ and is expressed as follows.
(5)$$\begin{array}{c} x_{u,j}^{r}=\left\{\begin{array}{c} 1,\mathrm{if}\;j=\arg\max_{j,r}\left\{\omega_{u,j}^{r}\right\},\\ 0,\mathrm{otherwise},\forall u\in U,\;\forall j\in J,\;\forall r\in R,\; \end{array}\right. \end{array}$$

We consider flat fading channels where all subchannels and channels over elements have similar channel gains. We also consider adaptive modulation and coding scheme (MCS) as in [10] with the SINR as in (Equation 4). Let Γ(.) denote the spectral efficiency obtained from MCS. Given SINR as in (Equation 4) and spectral efficiency Γ(.) (bits/symbol). If the user *u* is associated with BS *j* via RIS *r*, then the link rate would be a function of $\omega_{u,j}^{r}$ and is given as follows.
(6)$$L_{u,j}^{r}=\frac{\Gamma \left(\omega_{u,j}^{r}\right)\;SC_{OFDM}\;SY_{OFDM}C}{T_{sc}},\;j\in j,\;c\in C\;.$$Given the link rate *r* as in the above equation, the actual data rate of the user *u* in the RIS system is expressed as
(7)$$\lambda_{u}=\sum_{j\in J}\sum_{r\in R}x_{u,j}^{r}\beta_{u,j}^{r}L_{u,j}^{r},\forall u\in U,$$
where, the user association $x_{u,j}^{r}$ as in (Equation 5), link rate $L_{u,j}^{r}$ as in (Equation 6), and $\beta_{u,j}^{r}$ is the user scheduling time fraction for user *u* by BS *j* via RIS *r*. The $\beta_{u,j}^{r}$ from BS *j* to user *u* via RIS *r* as in [10] is given as
(8)$$\beta_{u,j}^{r}=\frac{{\left(L_{u,j}^{r}\right)}^{\frac{1-\alpha }{\alpha }}}{\sum_{u\in U}\sum_{r\in R}x_{u,j}^{r}{\left(L_{u,j}^{r}\right)}^{\frac{1-\alpha }{\alpha }}},\;\forall j\in J,$$
where, α is the fairness parameter, and $x_{u,j}^{r}$ is the user association between the BS *j* and user *u* via RIS *r* as in (Equation 5). Further, at any instant of time, only one user is served by the BSs for $\beta_{u,j}^{r}$ fraction of time. Similar to RIS channel parameters, the link rates, resultant rates, and the scheduling of a direct link between the BS *j* and the user *u* are expressed as $L_{u,j}^{0}$, $\lambda_{u}$, and $\beta_{u,j}^{0}$, respectively. These are computed based on the direct link association, channel gains, and association values as in (Equation 1)–(Equation 3), respectively. The performance metrics considered in this paper are presented next.

### 2.3. Performance Metrics

We consider coverage and α-fair throughput as our performance metric in this work. The coverage is defined as the probability of a user receiving SINR greater than the minimum threshold as in [10]. We consider α-fairness in this work. The utility function of $\Lambda_{\alpha }\left(x\right)$ with respect to general variable *x* is written as follows.
(9)$$\Lambda_{\alpha }\left(x\right)=\left\{\begin{array}{cc} \frac{x^{1-\alpha }}{1-\alpha }, & \alpha >0,\;\alpha \neq 1,\\ log(x) & \alpha =1, \end{array}\right.$$
where, α is the fairness parameter. When α=1, (Equation 9) leads to proportional fairness. As α increases, the worst case user’s rate of the system improves. α→∞ corresponds to max–min fairness where then minimum rate of the user in the system is maximized. The optimization problem formulated for time based RIS sharing is presented in the next section.

## 3. Time-Based RIS Sharing

We define time-based RIS sharing as the resource allocation of RIS in terms of time when shared among multiple BSs. The illustration of time-based RIS sharing between two BSs is shown in Figure 2 for two time slots t1 and t2, respectively. We define $x_{u,j}^{r}$ as a binary association variable that indicates the association between the user *u* and BS *j*. *r* indicates whether the association is direct or via RIS. If r=0, then it is direct association, otherwise, the user *u* is associated to BS *j* via RIS *r*. $A_{u,j}^{r}$ and the $D_{u,j}^{r}$ are the possible user scheduling time fractions by BS and RIS, respectively. We need to find the user scheduling time fraction that falls between $A_{u,j}^{r}$ and $D_{u,j}^{r}$. Given the total number of RIS elements (*N*) and the link rate $L_{u,j}^{r}$ as in (Equation 6), the user scheduling and the resource allocation of RIS in terms of time is formulated as follows.
(10)$$\begin{array}{ccc} P1 & : & \max_{\begin{array}{c} x_{u,j}^{r},\beta_{u,j}^{r} \end{array}}\sum_{u\in U}\Lambda_{\alpha }\left(\lambda_{u}\right), \end{array}$$
(11)s.tλu=∑r∈R∑j∈Jxu,jrβu,jrLu,jr(N,hu,jr),(12)∑u∈U∑r∈RAu,jr≤1,∀j∈J,(13)∑u∈U∑j∈JDu,jr≤1,∀r∈R,(14)βu,jr≤{Au,jr,Du,jr},(15)xu,jr∈{0,1},∀u∈U,∀j∈J,∀r∈R,(16)βu,jr∈[0,1],∀u∈U,∀j∈J,∀r∈R,(17)Au,jr∈[0,1],∀u∈U,∀j∈J,(18)Du,jr∈[0,1],∀u∈U,∀j∈J,∀r∈R,
where, the sum of α−fair utility function as in (Equation 9) for all the users in the system is given in (Equation 10). The resultant data rate of the user *u* is specified in (Equation 11). The constraint on (12) and (13) specifies that the sum of total time fraction allocated by BS *j* and the RIS *r* should be less than or equal to 1, respectively. Based on $A_{u,j}^{r}$ and $D_{u,j}^{r}$, the actual user scheduling time fraction $\beta_{u,j}^{r}$ in (Equation 11) is computed over the set of BSs associated to the RIS *j* and this is given in (14). The binary constraint on the association variable $x_{u,j}^{r}$ is given in (15). The constraints on (16)–(18) give the continuous range between [0, 1] of $\beta_{u,j}^{r}$, $A_{u,j}^{r}$ and $D_{u,j}^{r}$ variables, respectively. Based on the above, the computed time fraction value should be between 0 and 1. The problem defined in (Equation 11) is mixed-integer nonlinear programming (MINLP) and is difficult to solve. The problem is decomposed into association and the scheduling sub-problems. The association is considered to be based on maximum received SINR as in (Equation 4). In order to solve for the optimized phase $\theta_{r,n}$ and the reflection coefficient $\phi_{r,n}$ of nth antenna element of rth RIS, we consider focusing principle of RIS. All the elements of RIS are assumed to focus on a single user at an instant of time. The evolution of state-of-the-art RIS hardware with higher programming capability is increasing nowadays. Thus, this focusing is considered to be possible technology with the RIS. Given the channel, $y_{j}^{r,n}$ and $z_{u,j}^{r,n}$ for SINR of the RIS link as in (Equation 4), the maximum rate is achieved by selecting $\theta_{r,n}=∠h_{u,j}^{0}-∠y_{j}^{r,n}z_{u,j}^{n}$ for each element *n* [12]. We assume a flat fading channel, i.e., the channel between BS *j* and every element *n* of the RIS *r* is the same. However, the proposed heuristics in this can be generalized to frequency selective fading channel by computing the channel matrix and given as input to the heuristic. Therefore, the channel between BS *j* and RIS *r* is now denoted as $y_{j}^{r}$. Similarly, the channel between RIS *r* and user *u* as $z_{u,j}^{r}$ and the amplitude of the reflection coefficient as 1.

The entire RIS is now focusing towards a single user *u*. The modified SINR with the channel and the optimal phase shifts can be conclusively written as in [12] as
(19)$$\omega_{u,j}^{r}=\frac{P_{j}{\left(\sqrt{h_{u,j}^{0}}+N\sqrt{y_{j}^{r}z_{u}^{r}}\right)}^{2}}{\sum_{j^{\prime }\in J\backslash j}P_{j^{\prime }}h_{u,j^{\prime }}^{0}+\sigma^{2}},\;\forall u\in U,\forall r\in R,r\geq 1,\;\;$$
where, $\sum_{j^{\prime }\in J\backslash j}P_{j^{\prime }}h_{u,j^{\prime }}^{0}$ is the interference received by the user *u* from other BSs, $P_{j}$ is the transmit power. Since the RIS parameters are optimized and the SINR is computed as in (Equation 19), the goal is now to determine the $\beta_{u,j}^{r}$ based on the computed SINR values. So, given the association $x_{u,j}^{r}$, we propose a heuristic Algorithm 1 to solve for $\beta_{u,j}^{r}$ of (Equation 11).
**Algorithm 1:** Proposed heuristic for time-based RIS sharing.1:INPUTS: $\left\{P_{j}^{r}h_{u,j}^{r}\right\}$, U2:OUTPUT: $\lambda_{u}$3:Initialize: u = 1, $\{x_{u,j}^{r}\}=0$4:**Repeat**5:Sort $\{P_{j}^{r}h_{u,j}^{r}\},\forall r\geq 1$ in decreasing order of received power from RIS. The rearranged order is represented using *q*. r=0 implies direct link between BS and user.6:**if**$P_{j}h_{u,j}^{r}\geq P_{j}h_{u,j}^{0}$**then**7:   $\omega_{u,j}^{r}=f\left(\left\{P_{j}h_{u,j}^{r}\right\}\right)$ as in (Equation 19)8:   $x_{u,j}^{r}=1$9:**else**10:   $\omega_{u,j}^{0}=f\left(\left\{P_{j}h_{u,j}^{0}\right\}\right)$ as in (Equation 3)11:   $x_{u,j}^{0}=1$12:**end if**13:   Set u=u+114:**Until**u>U15:Set u = 116:**Repeat**17:   Compute $\left(A_{u,j}^{r}\right)$ as in (Equation 8)18:   Compute $\left(D_{u,j}^{r}\right)$ as in (Equation 8)19:   Compute $\beta_{u,j}^{r}=min\left\{A_{u,j}^{r},D_{u,j}^{r}\right\}$20:   Compute $\lambda_{u}$ as in (12)21:   Set u=u+122:**Until**u>U23:**Stop**

The heuristic in Algorithm 1 starts with the set of received powers, channel parameters, and the set of users as inputs. The output is the resultant data rate of each user with its optimal user scheduling time fractions. The overall time-based RIS sharing heuristic is divided into two parts. The first part, Steps 3–14, considers the user association and the RIS user selection based on the available received powers and the channel parameters. Once the users are classified as a direct link and the RIS link users, the second part of the heuristic computes the appropriate user scheduling non-overlapping time fractions of all the RIS users from all the participating BSs in the system. Based on the available user information from the BS *j*, the BS first computes user *u*’s scheduling time fraction $A_{u,j}^{r}$ as in (Equation 8). Similarly, based on the overall RIS users information from different BSs at the RIS, the $D_{u,j}^{r}$ is computed as in (Equation 8). For all the RIS users, the minimum of $A_{u,j}^{r}$ and $D_{u,j}^{r}$ is considered to be user *u*’s scheduled time fraction from BS *j* via RIS *r*. The RIS has only limited resources, and the total time fraction in BSs and the shared RIS should on exceed 1 as in (13) and (14), respectively. Suppose any one of $A_{u,j}^{r}$ and $D_{u,j}^{r}$ is selected randomly as the time fraction without considering the constraints. In that case, there will be inconsistency and overlapping in their time allocation. Thus, this minimum of $A_{u,j}^{r}$ and $D_{u,j}^{r}$ is considered to be the effective way to schedule the users when RIS is shared among multiple BSs. The computation of $D_{u,j}^{r}$ can be RIS-centric (RIS computes $D_{u,j}^{r}$) or BS-centric (BS computes $D_{u,j}^{r}$). The proposed heuristic works irrespective of whether the computation is at RIS or the BS. However, there is a trade-off between RIS- and BS-centric computation. If the $D_{u,j}^{r}$ can be computed at the centralized BS where the RIS users’ information of all participating BSs is shared, then the complexity at the RIS is reduced. On the contrary, if $D_{u,j}^{r}$ is computed at the RIS, then the information overhead at the BS is reduced. However, in both the cases, BS *j* decides $\beta_{u,j}^{r}$ for its user *u*. Further, the priority of the BSs that need RIS access is decided either by RIS (RIS-centric) or by the centralized BS (BS-centric). This prioritizing is required to avoid allocating the same time slots to two or more BSs. Next, we present the problem formulation of element-based RIS sharing.

## 4. Element Based RIS Sharing

In this section, we formulate and propose a heuristic for resource allocation of RIS elements when a RIS is shared among multiple BSs. An example of RIS sharing in terms of elements between two BSs is illustrated in Figure 3. We define $\eta_{j}^{r}$ as the fraction of elements of RIS *r* allocated to BS *j*. The SINR computed as in (Equation 19) is the function of the number of elements of RIS allocated to a particular BS *j*. Therefore, the link rate of user *u* is the function of the number of RIS elements allocated to particular BS *j*, i.e., the number of RIS elements allocated to BS *j* is equal to the fraction of RIS elements allocated multiplied by the total number of RIS elements. Given the binary association $x_{u,j}^{r}$ as before, the resource allocation of RIS elements and the user scheduling time fraction are formulated as below.
(20)$$\begin{array}{ccc} P2 & : & \max_{\begin{array}{c} \beta_{u,j}^{r},\eta_{j}^{r} \end{array}}\sum_{u\in U}\Lambda_{\alpha }\left(\lambda_{u}\right), \end{array}$$
(21)s.tλu=∑r∈R∑j∈Jxu,jrβu,jrLu,jr(ηjr,N,hu,jr),(22)∑u∈U∑r∈Rβu,jr≤1,∀j∈J,(23)∑j∈Jηjr≤1,∀r∈R,(24)xu,jr∈{0,1},∀u∈U,∀j∈J,∀r∈R,(25)ηjr∈[0,1],∀j∈J,∀r∈R,(26)βu,jr∈[0,1],∀u∈U,∀j∈J,∀r∈R,
where, the objective function in (Equation 20) specifies the sum of the α−fair utility function. The resultant rate of user *u* is computed using (Equation 21). Equation (Equation 21) is a function of fraction of RIS elements allocated to particular BS *j* ($\eta_{j}^{r}$). The sum of user scheduling time fraction in any BS *j* should not exceed 1, and this is given ion constraint (22). Equation (23) specifies that the sum of the fraction of elements of RIS allocated to BSs should not exceed 1. The binary association constraint is specified on (24). Equations (25) and (26) specify the positivity constraints of $\eta_{j}^{r}$ and $\beta_{u,j}^{r}$, respectively. The problem in (Equation 20) is a discrete optimization problem where (Equation 21) depends primarily on $\eta_{j}^{r}$, and this (Equation 21) is computed for different values of $\eta_{j}^{r}$ using MCS in [10]. In order to compute the optimal value of $\eta_{j}^{r}$, the objective function in (Equation 20) is now computed for 10 discrete values in the set as ϵ={0.1,0.2,…1}. The heuristic for computing (Equation 20) using ϵ is presented in Algorithm 2.

The set of fraction of RIS elements ({ϵ}), received powers ($\left\{P_{j}^{r}h_{u,j}^{r}\right\}$), and the set of users (U) is given as input to the heuristic. The resultant user rates and the sum of its utility functions are computed for all ϵ values. The maximum of those fractions is chosen as the fraction of RIS elements allocated to a particular BS. For a system with 2 BS, there are 11 possibilities for allocating RIS elements. When three BSs share a single RIS, the number of computations increases to 66. Considering the BSs in proximity are not more than 3 or 4 at any given RIS location, the complexity of the heuristic for computing the fraction of RIS elements required for a single BS is considerable. Similar to time based RIS sharing, the heuristic in Algorithm 2 is also independent of RIS- or BS-centric computation. Therefore, the fraction of RIS elements $\eta_{j}^{r}$ allocated to each BS *j* can be computed at the RIS (RIS-centric) or the centralized BS (BS-centric). The numerical results corresponding to the proposed methods are discussed next.
**Algorithm 2:** Proposed heuristic for element-based RIS sharing.1:INPUTS: $\left\{P_{j}^{r}h_{u,j}^{r}\right\}$, U, ϵ={0.1,0.2,…,1}2:OUTPUT: $\lambda_{u}$, $\eta_{j}^{r}$3:Initialize: u = 1, $\{x_{u,j}^{r}\}=0$, q=14:**Repeat**5:**Repeat**6:Sort $\{P_{j}^{r}h_{u,j}^{r}\},\forall r\geq 1$ in decreasing order of received power from RIS. The rearranged order is represented using *q*. r=0 implies direct link between BS and user.7:**if**$P_{j}h_{u,j}^{r}\geq P_{j}h_{u,j}^{0}$**then**8:   $\omega_{u,j}^{r}=f\left(\left\{P_{j}h_{u,j}^{r}\right\}\right)$ as in (Equation 19)9:   $x_{u,j}^{r}=1$10:**else**11:   $\omega_{u,j}^{0}=f\left(\left\{P_{j}h_{u,j}^{0}\right\}\right)$ as in (Equation 3)12:   $x_{u,j}^{0}=1$13:**end if**14:   Set u=u+115:**Until**u>U16:Set u = 117:**Repeat**18:   Compute $\beta_{u,j}^{r}$ as in (Equation 8)19:   Compute $\lambda_{u}$ as in (Equation 21)20:   Set u=u+121:**Until**u>U22:Compute $\chi_{q}=\sum_{u\in U}\Lambda_{u}$, $\left\{\varphi_{q}\right\}=\left\{\lambda_{1},\lambda_{2},\ldots ,\lambda_{U}\right\}$23:**Until**q>|ϵ|24:$q^{*}=\max_{q}\left\{\chi_{1},\chi_{2},\cdots ,\chi_{q}\right\}$25:${\left(\eta_{j}^{r}\right)}^{*}=\epsilon \left(q^{*}\right)$, $\lambda_{u}=\left\{\varphi_{q^{*}}\right\}$26:**Stop**

## 5. Numerical Results

We consider area of 200m×200m with the RIS location at (0, 10). The BSs, for instance, are located at (−100, 0) and (100, 0). An snapshot of simulation setup with the considered area is shown in Figure 4. Table 2 specifies the set of simulation parameters and channel model. The total bandwidth used in this simulation is equal to subchannel bandwidth ×C. The users are distributed uniformly and randomly in the given area with the density $\mu_{u}$. We also consider $\mu_{b}$ as the blocker density in the given area. We also consider a set of users in the blocked area or circled area near RIS as in Figure 4. The density of the users in the circled or blocked area $\mu_{pu}$ varies from 20–80% of $\mu_{u}$ at a given instant of time, and the same amount of users are removed from the original distribution of the considered area to maintain the density $\mu_{u}$. This method is useful to leverage the advantage of RIS and RIS sharing scenarios. The users are associated to BSs based on maximum received power as in (Equation 1) for the direct link and (Equation 5) for the RIS link as given in the heuristic Algorithms 1 and 2. The throughput equivalent to (Equation 9) is given as follows [10].
(27)$$T_{\alpha }=\left\{\begin{array}{cc} \;{\left(\frac{1}{U}\sum_{u\in U}\lambda_{u}^{1-\alpha }\right)}^{\frac{1}{1-\alpha }} & ,\alpha >0,\;\alpha \neq 1,\\ \;{\left(\prod_{u\in U}\lambda_{u}\right)}^{\frac{1}{U}} & ,\alpha =1, \end{array}\right.$$
where, α is the fairness parameter, $\lambda_{u}$ is as defined in (12) and (Equation 21) for time and element based RIS sharing, respectively. The throughput and coverage are calculated and averaged over $10^{5}$ realizations for different user and blocker densities. For each location realization, the fading is averaged over 100 iterations. We present results for fairness parameter α=1 in this paper. However, the formulated problem and the proposed heuristics can be generalized to any fairness parameter α.

The trade-off between coverage and throughput for α=1, user density of 40, blocker density of 40, and the number of RIS elements *N* as 40 is shown in Figure 5. It is observed from the Figure 5 that the proposed time based RIS sharing performs better in terms of throughput and coverage for the given scenario than the other schemes. Although element-based RIS sharing performs better than the benchmark and the RIS without sharing schemes, it has a slight performance degradation than the time based RIS sharing. This slight performance decrease is because all the elements of RIS focus towards a single user at any point of time. However, element based RIS sharing has only a particular fraction of RIS elements focused towards a user.

The variation of throughput with respect to the user densities $\mu_{u}$ is shown in Figure 6. The time- and element-based RIS sharing outperform the benchmark without sharing schemes. The throughput of the proposed methods decreases with the increase in the user densities $\mu_{u}$. The resources available to the system are limited, and when the number of users increases, the resources allocated to these users decreases. Thus, the throughput for all the schemes decreases as the user density increases. We can infer the same scenario with the varying percentage of users in blocked area $\mu_{pu}$ as shown in Figure 7.

The throughput variation with respect to the blocker density $\mu_{b}$ in the given area is shown in Figure 8. As $\mu_{b}$ increases, more users lose coverage, and hence throughput decreases in Figure 8. However, the proposed time- and element-based heuristic performs better than the other two schemes with varying $\mu_{b}$. The variation of throughput with respect to the number of RIS elements (*N*) for all the schemes is shown in Figure 9. As the number of RIS elements *N* increases, the proposed time- and element-based RIS sharing methods perform better than benchmark and RIS without sharing schemes. The difference in throughput between time- and element-based RIS sharing is almost negligible when N increases. For example, as in Figure 9, there is an increase of 14% throughput for time-based RIS sharing when compared to element-based RIS sharing. However, this decreases to 1.6%, and 0.9% as *N* increases to 500 and 1000, respectively. We proposed both time- and element-based RIS sharing in this paper. Although the element-based RIS sharing performs slightly lesser than the time based RIS sharing in the given scenario, we expect that the element-based RIS sharing would be helpful when multiple operators share a single RIS or scenarios with high user mobility. The configuration and the allocation of RIS elements in a element-based RIS sharing are more manageable during the above scenarios at the expense of a slight decrease in the throughput. *Overall, the proposed time and element based RIS sharing performs better in terms of coverage and throughput with respect to user densities ($\mu_{u}$), percentage of users in blocked area ($\mu_{pu}$), blocker densities ($\mu_{b}$), and the number of RIS elements (N).*

## 6. Conclusions

The resource allocation of RIS sharing among multiple BSs in terms of time and RIS elements are formulated as an optimization problem. Since the formulated problems are difficult to solve, two heuristics are proposed to solve the time- and element-based RIS sharing. The throughput and coverage results are presented for different user and blocker densities. The proposed time- and element-based RIS sharing performs better in terms of coverage and throughput when compared to the benchmark and the RIS without sharing schemes. For specific scenarios, the proposed time- and element-based RIS sharing schemes show a 53% and 25% increase in throughput, respectively, when compared to RIS system without sharing. Although the element-based RIS sharing performs slightly lesser than the time based RIS sharing, the throughput of element-based RIS sharing is considerably higher than the benchmark and RIS without sharing schemes. Further, we consider exploring suitable scenarios for element-based RIS sharing, where it can be advantageous in terms of coverage and throughput.

## Figures and Tables

**Figure 1 sensors-22-05619-f001:**
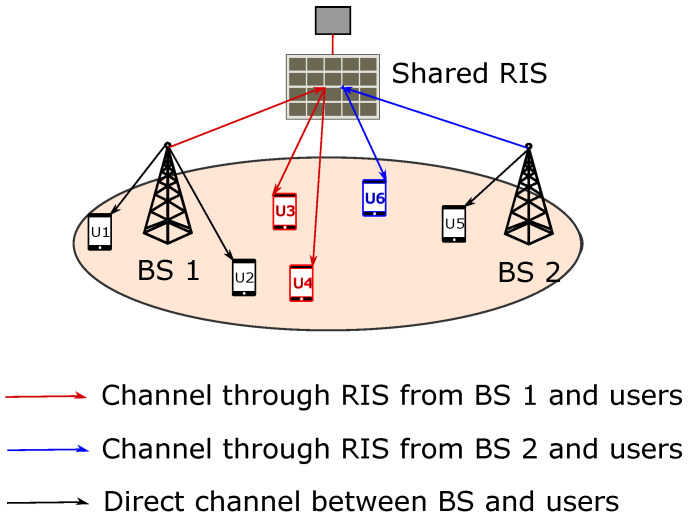
System model of RIS sharing.

**Figure 2 sensors-22-05619-f002:**
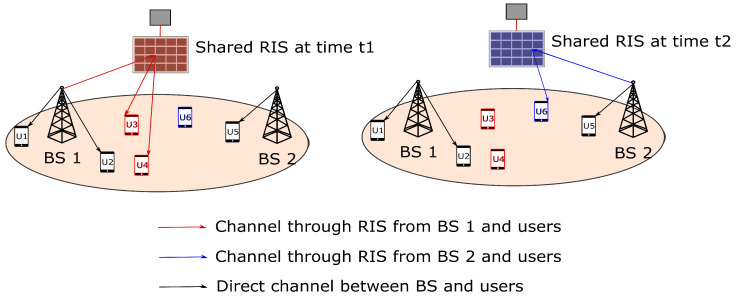
System model of time based RIS sharing.

**Figure 3 sensors-22-05619-f003:**
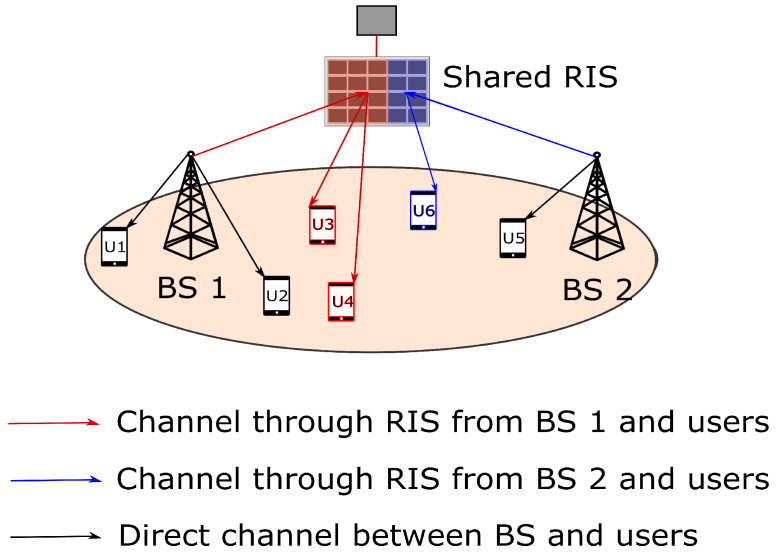
System model of element-based RIS sharing.

**Figure 4 sensors-22-05619-f004:**
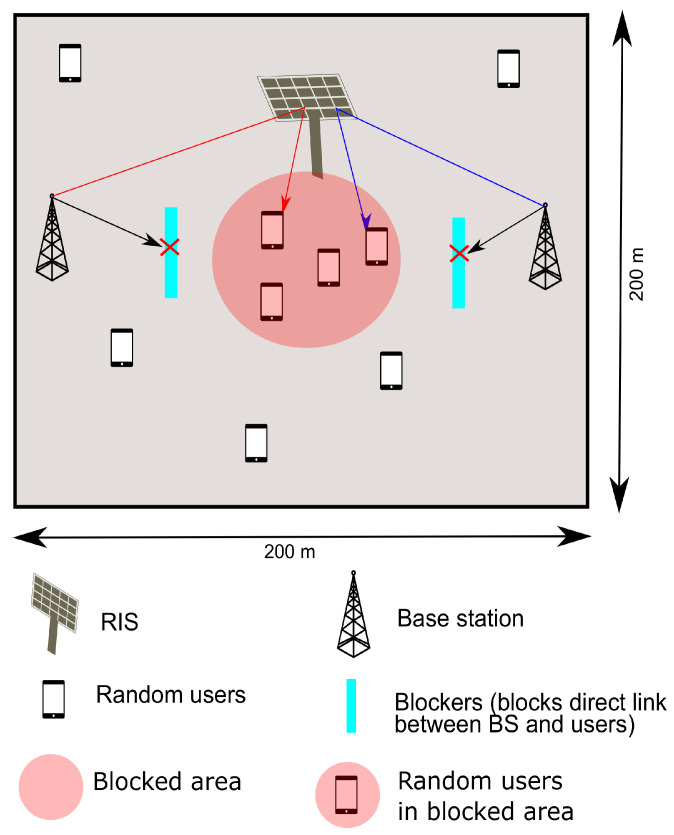
RIS sharing simulation setup.

**Figure 5 sensors-22-05619-f005:**
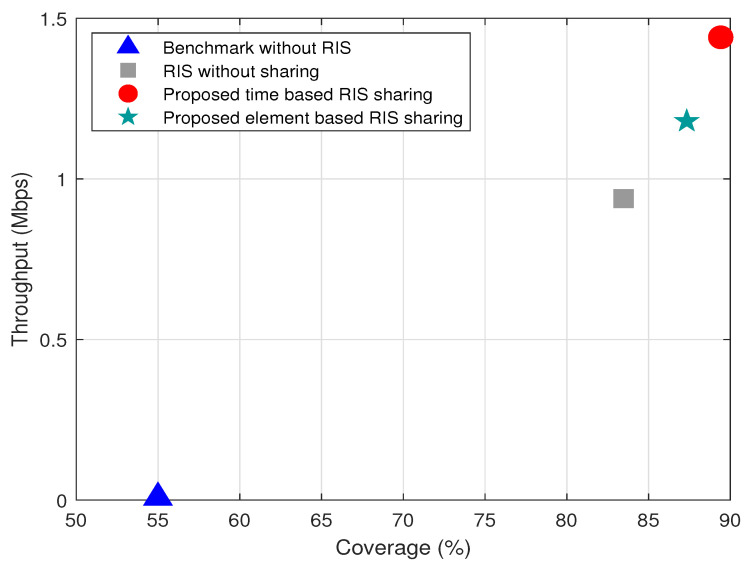
Variation of throughput with respect to coverage for $\mu_{u}=40$, $\mu_{pu}=40\%$, N=40, $\mu_{b}=2$, and α=1.

**Figure 6 sensors-22-05619-f006:**
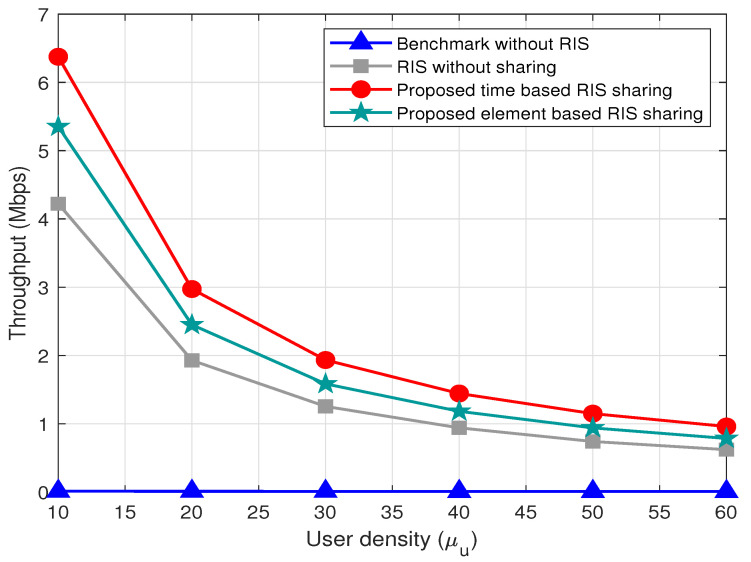
Variation of throughput with respect to $\mu_{u}$ for $\mu_{pu}=40\%$, N=40, $\mu_{b}=2$, and α=1.

**Figure 7 sensors-22-05619-f007:**
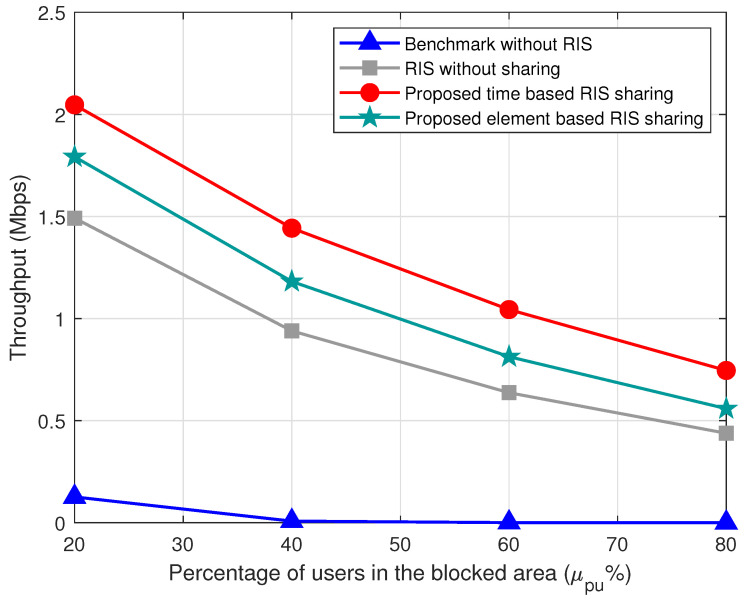
Variation of throughput with respect to $\mu_{pu}$ users in the blocked area for $\mu_{u}=40$, N=40, $\mu_{b}=2$, and α=1.

**Figure 8 sensors-22-05619-f008:**
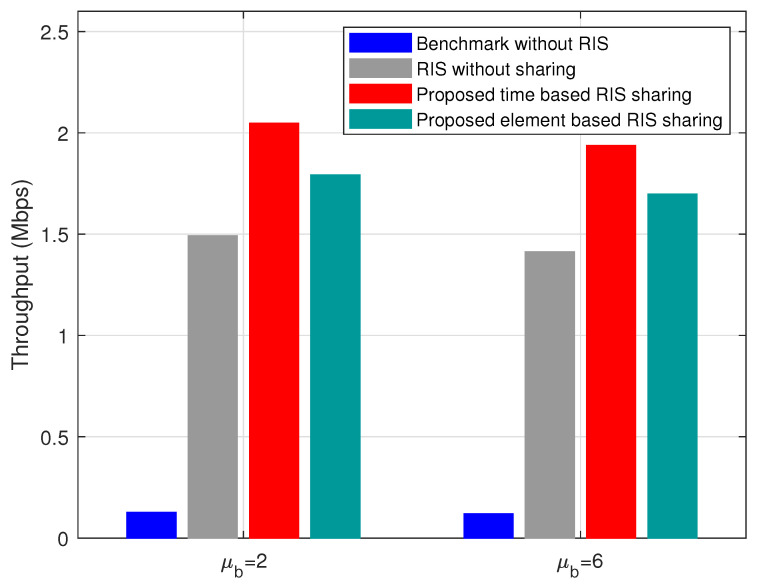
Variation of throughput with respect to blocker density $\mu_{b}$ for $\mu_{pu}=20\%$, N=40, $\mu_{b}=2$, and α=1.

**Figure 9 sensors-22-05619-f009:**
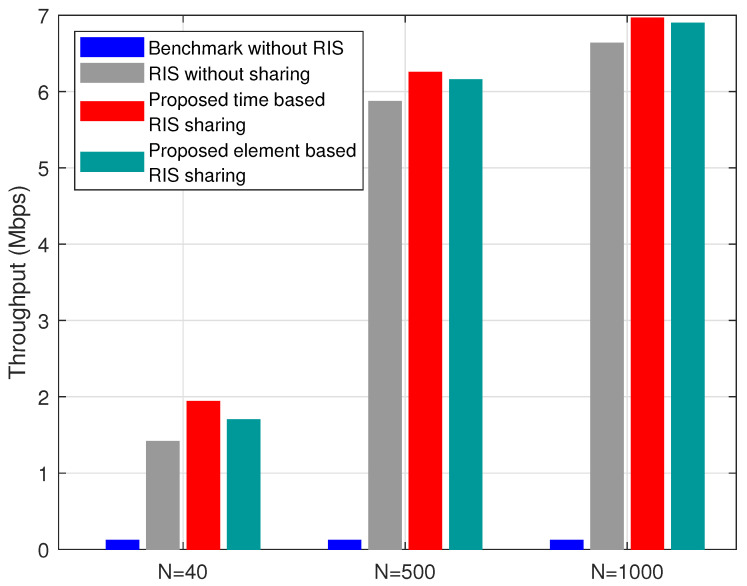
Variation of throughput with respect to number of of RIS elements *N* for $\mu_{u}=40\%$, $\mu_{pu}=20\%$, $\mu_{b}=6$, and α=1.

**Table 1 sensors-22-05619-t001:** Mathematical notations.

$h_{u,j}^{0}$	Direct channel gain at user *u* from BS *j*
${\hat{h}}_{u,j}^{r}$	Channel gain at user *u* from BS *j* via RIS *r*
$L_{u,j}^{0}$	Direct link rate of user *u* from BS *j*
$L_{u,j}^{r}$	Link rate of user *u* from BS *j* via RIS *r*
N	Number of elements in RIS
$x_{u,j}^{0}$	Binary user association variable of user *u* with BS *j* without RIS
$x_{u,j}^{r}$	Binary user association variable of user *u* with BS *j* via RIS
$y_{j}^{r,n}$	Channel between BS *j* and nth element of RIS *r*
$z_{u,j}^{r,n}$	Channel between BS *j* and user *u* via nth element of RIS *r*
α	Fairness parameter for the α-Fair scheduler
$\beta_{u,j}^{0}$	User scheduling time fraction for user *u* by BS *j* without RIS
$\beta_{u,j}^{r}$	User scheduling time fraction for user *u* by BS *j* via RIS *r*
Γ(.)	Spectral efficiency in bits/symbol
$\omega_{u,j}^{0}$	DL received SINR of user *u* from a BS *j*
$\omega_{u,j}^{r}$	DL received SINR of user *u* from a BS *j* via RIS *r*
$\mu_{b}$	Number of blockers
$\mu_{u}$	Number of users
$\mu_{pu}$	Number of users in the blocked area
$\Lambda_{\alpha }$	α-fair utility function
$\lambda_{u}$	Resultant data rate of user *u*
$T_{\alpha }$	α-fair throughput

**Table 2 sensors-22-05619-t002:** Simulation parameters.

$f_{c}$	28 GHz
Area	200 × 200 m
Penetration loss ($\upsilon_{sc}$)	20 dB for NLOS path
Loss due to shadowing (ρ)	Standard deviation of 4 dB
P${}_{j}$	35 dBm
PL(d)	Urban micro [13]
*C*	99
Subchannel Bandwidth	720 KHz
SC${}_{OFDM}$	12
SY${}_{OFDM}$	14
T${}_{Subframe}$	0.25 ms

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
