# Peer review of "Resource Allocation and Sharing Methodologies When Reconfigurable Intelligent Surfaces Meet Multiple Base Stations"

_sensors, 2022, doi:10.3390/s22155619_

Round 1

Reviewer 1 Report

The paper addresses a relevant topic such as 5G and intelligent surfaces. The descriptions are exhaustive, inclusing mathematical expressions and several details within the text. The proposal in technically sound, as results support the conclusions and the initial claims are finally validated through different experiements. A comparison to the state of the art mechanisms is also provided. In my opinion the paper may be accepted

Author Response

Resource Allocation And Sharing Methodologies When Reconfigurable Intelligent Surfaces Meet Multiple Base Stations

sensors-1818452

Yoghitha Ramamoorthi * , Riku Ohmiya , Masashi Iwabuchi , Tomoaki Ogawa , Yasushi Takatori

Dear Reviewer 1

Thank you for your kind comments and suggestions. Please find our responses below.

All reviewers’ comments in this document are written in red. 

Response to Reviewer1’s Comments

Point 1: The paper addresses a relevant topic such as 5G and intelligent surfaces. The descriptions are exhaustive, including mathematical expressions and several details within the text. The proposal is technically sound, as results support the conclusions and the initial claims are finally validated through different experiements. A comparison to the state of the art mechanisms is also provided. In my opinion the paper may be accepted

Response 1: Thank you for your kind comments. We have made minor revisions to the manuscript based on the other reviews. We hope that you will be satisfied with the revised version of this manuscript.

Reviewer 2 Report

This paper has well defined and well organized. However, it would be better if the author plotted the spectral efficiency curve, bit error rate vs. signal-to-noise ratio with different modulation schemes.

Author Response

Resource Allocation And Sharing Methodologies When Reconfigurable Intelligent Surfaces Meet Multiple Base Stations

sensors-1818452

Yoghitha Ramamoorthi * , Riku Ohmiya , Masashi Iwabuchi , Tomoaki Ogawa , Yasushi Takatori

Dear Reviewer 2

Thank you for your kind comments and suggestions. Please find our responses below.

All reviewers’ comments in this document are written in black and their responses are written in red. The text in blue color shows the updated portion of the revised manuscript.

Response to Reviewer 2’s Comments

Point 1: This paper has well defined and well organized. However, it would be better if the author plotted the spectral efficiency curve, bit error rate vs. signal-to-noise ratio with different modulation schemes.

Response 1: Thank you for your kind comments. The results with spectral efficiency, bit error rate vs signal-to-noise ratio with different modulation schemes would be better to understand from the perspective of the link level simulations. However, we focused on system level performance metrics such as system throughput and coverage in this manuscript. The system level performance of the different methodologies of RIS sharing has been presented in the manuscript.

Point 2: In page 7: Name of Figure 3, will be System Model of element based RIS Sharing

Response 2: Thank you for your kind comment.

  • On page 7, name of Figure 3 is changed to, “System Model of element based RIS Sharing”

We have made minor revisions to the manuscript based on the other reviews. We hope that you will be satisfied with the revised version of this manuscript.

Reviewer 3 Report

1. There are semantic and grammatical errors in the expressions in the paper.

2. Some formulas in the text are incorrectly formatted and lack punctuation marks. When explaining parameters, commas should not be added after where.

3. The explanation of the parameters and symbols in the equation is not comprehensive enough, and an explanation table of symbols and parameters can be added.

4. The scene graph in the paper is single and does not indicate the actual application scene, such as urban environment or mountain environment. The lack of obstacles such as buildings and trees does not make it clear that the direct link is unreachable.

5. The explanation of the scene figure in the paper is not clear and the explanation is insufficient.

6. The meanings expressed in Figure 2 and Figure 3 in the text are redundant, and Figure 2 can be deleted after being marked on Figure 3.

7. Edge Computing or networking techniques for promoting 6G should be further discussed. The following work should be discussed for considering their advances and feasibilities for RIS.

Liang Zhao, Weiliang Zhao, Ammar Hawbani, Ahmed Al-Dubai, Geyong Min, Albert Y. Zomaya, Changqing Gong, “Novel Online Sequential Learning-based Adaptive Routing for Edge Software-Defined Vehicular Networks,” IEEE Transactions on Wireless Communications, 2020

8. The structure of the article in the fifth chapter of the paper is not clear, and the second algorithm is inappropriate in this chapter, and the article structure can be readjusted.

After the algorithm part is integrated, a single chapter will be opened.

Author Response

Resource Allocation And Sharing Methodologies When Reconfigurable Intelligent Surfaces Meet Multiple Base Stations

sensors-1818452

Yoghitha Ramamoorthi * , Riku Ohmiya , Masashi Iwabuchi , Tomoaki Ogawa , Yasushi Takatori

Dear Reviewer 3

Thank you for your kind comments and suggestions. Please find our responses below.

All reviewers’ comments in this document are written in black and their responses are written in red. The text in blue color shows the updated portion of the revised manuscript.

Response to Reviewer 3’s Comments

Point 1: There are semantic and grammatical errors in the expressions in the paper.

Response 1: Thank you for your kind comment. Based on your comment. We have thoroughly checked our manuscript and made changes appropriately.

Point 2: Some formulas in the text are incorrectly formatted and lack punctuation marks. When explaining parameters, commas should not be added after where.

Response 2: Thank you for your kind comment. Based on your comment, we have checked the equation’s punctuation marks and its explaining parameters on our manuscript. The changes are made in the manuscript accordingly.

Point 3: The explanation of the parameters and symbols in the equation is not comprehensive enough, and an explanation table of symbols and parameters can be added.

Response 3: Thank you for your kind comments. Based on your comment, we have added a table of symbols and notations used in this manuscript. The details of the table added in the manuscript are given below.

  • On page 3 of the revised manuscript, “Table 1: named Mathematical Notations” has been added.

Point 4: The scene graph in the paper is single and does not indicate the actual application scene, such as urban environment or mountain environment. The lack of obstacles such as buildings and trees does not make it clear that the direct link is unreachable.

Response 4: Thank you for your kind comment. We considered urban micro with the transmit power of 30dBm as mentioned in Table 2 of the revised manuscript.  We defined the blockage as an obstacle in general (can be a tree or building) with a specific height and width. If a blockage exists between the user $u$ and BS $j$, then the direct link is assumed to be blocked and there is no line of sight (LoS) link between the user $u$ and BS $j$.

  • On page 10 of the revised manuscript, Figure 4 has been slightly modified to show that there is no LoS between BS and the users.

Point 5: The explanation of the scene figure in the paper is not clear and the explanation is insufficient.

Response 5: Thank you for your kind comment. The first paragraph of Section , Numerical section gives an explanation of the scene Fig. 4.

  • On page 10, Section 5, Numerical Results, a line has been added for better understanding, “A snapshot of the simulation setup with the considered area is shown in Fig. 4.”

Point 6: The meanings expressed in Figure 2 and Figure 3 in the text are redundant, and Figure 2 can be deleted after being marked on Figure 3.

Response 6: Thank you for your kind comment. The name of Figure. 3 has been modified in the revised manuscript. 

  • On page 7 of the revised manuscript, the name of Figure 3 has been changed to, “System Model of element based RIS Sharing”

Point 7: Edge Computing or networking techniques for promoting 6G should be further discussed. The following work should be discussed for considering their advances and feasibilities for RIS.

Liang Zhao, Weiliang Zhao, Ammar Hawbani, Ahmed Al-Dubai, Geyong Min, Albert Y. Zomaya, Changqing Gong, “Novel Online Sequential Learning-based Adaptive Routing for Edge Software-Defined Vehicular Networks,” IEEE Transactions on Wireless Communications, 2020

Response 7: Thank you for your kind comment. The mentioned reference has been added and the details are given below.

  • On page 14 of the revised manuscript, the new reference has been added as Reference [11].
  1. Zhao et al. Novel Online Sequential Learning-Based Adaptive Routing for Edge Software-Defined Vehicular Networks. IEEE Trans. on Wire. Comm, 2021, 20, 2991–300.
  • On page 2 of the revised manuscript, A line has been added for reference[11], “The prospective use cases of 6G includes RIS assisting wireless networks [1], NOMA [3], and edge computing [11], etc., as in [4]-[10]”.

Point 8: The structure of the article in the fifth chapter of the paper is not clear, and the second algorithm is inappropriate in this chapter, and the article structure can be readjusted.

After the algorithm part is integrated, a single chapter will be opened.

Response 8: Thank you for your comments. This is due to the wrong positioning of Algorithm. 2 in Section. 5. We modified this in the revised manuscript. The proposed Algorithm. 2 belongs to Section 4. However, please find the responses in detail.

  • Point 8-1: The structure of the article in the fifth chapter of the paper is not clear

Response 8-1: The fifth chapter (Section 5) contains Numerical results of the proposed system along with the simulation parameters. This starts with the description of the scenario considered and other parameters of the simulation. The trade-off between coverage and throughput of the system is presented first. It is followed by the variation of throughput with respect to the number of users, number of users in the blocked area, number of blockers, and the number of RIS elements (N), respectively.

  • Point 8-2: the second algorithm is inappropriate in this chapter

Response 8-2: Thank you for your kind comment. We understood the confusion because of the positioning of Algorithm. 2 (Element based RIS sharing). It is nopositioneded appropriately in Section. 4 Element based RIS sharing.

  • On page 9 of the revised manuscript, Algorithm. 2 is now placed in Section. 4.

  • Point 8-3: the article structure can be readjusted.

Response 8-3: Regarding the article structure, we started with the  Introduction of RIS and related works. The system model and the performance metrics considered are explained in the next section. The proposed time and element based RIS sharing and the methodologies in Section. 3 (Algorithm 1) and Section 4 (Algorithm 2), respectively. After addressing Point 8-2, the position of Algorithm. 2 would be clearer.

We have made minor revisions to the manuscript based on the other reviews. We hope that you will be satisfied with the revised version of this manuscript.
